# Extending the Usage of Newton's Method with Applications to the Solution of Bratu's Equation

**Ioannis K. Argyros** [1,†] and **Daniel González** [2,*,†]

1 Department of Mathematical Sciences, Cameron University, Lawton, OK 73505, USA; iargyros@cameron.edu
2 Escuela de Ciencias Físicas y Matemáticas, Universidad de Las Américas, Quito 170125, Ecuador
* Correspondence: daniel.gonzalez.sanchez@udla.edu.ec; Tel.: +593-2-3970-000
† These authors contributed equally to this work.

**Abstract:** We use Newton's method to solve previously unsolved problems, expanding the applicability of the method. To achieve this, we used the idea of restricted domains which allows for tighter Lipschitz constants than previously seen, this in turn led to a tighter convergence analysis. The new developments were obtained using special cases of functions which had been used in earlier works. Numerical examples are used to illustrate the superiority of the new results.

**Keywords:** Bratu; Newton; convergence

## 1. Introduction

Let $F : \Omega \subset E_1 \to E_2$ be a differentiable operator in the sense of Fréchet, where $E_1$ and $E_2$ are Banach spaces and $\Omega$ is a nonempty and open set. A plethora of problems from many diverse disciplines are formulated using modeling which looks like

$$F(x) = 0. \tag{1}$$

Hence, the problem of locating a solution $x_*$ for Equation (1) is very important. Most people develop iterative algorithms approximating $x_*$ under some conditions, since a closed form solution cannot easily be obtained in general. The most widely used iterative method is Newton's method defined for an initial point $x_0 \in \Omega$ by

$$\begin{cases} x_0 \in \Omega, \\ x_{n+1} = x_n - F'(x_n)^{-1} F(x_n) \quad \text{for all} \quad n = 0, 1, 2 \dots \end{cases} \tag{2}$$

Numerous convergence results appear in the literature based on which $\lim_{n \to +\infty} x_n = x_*$.

In this article, we introduce new semilocal convergence results based on our idea of restricted convergence region through which we locate a more precise set containing $x_n$. This way, the majorizing constants and scalar functions are tighter leading to a finer convergence analysis.

To provide the semilocal convergence analysis Kantorovich used the condition [1]

$$\|F''(x)\| \leq k, \quad x \in \Omega. \tag{3}$$

Let function $\psi : \mathbb{R}_+ \cup \{0\} \to \mathbb{R}$ be non-decreasing and continuous. A weaker condition is [2–6]

$$\|F''(x)\| \leq \psi(\|x\|), \quad x \in \Omega. \tag{4}$$

We shall find a tighter domain than $\Omega$, where Equation (4) is satisfied. This way the new convergence analysis shall be at least as precise.

The layout of the rest of the article involves the semilocal convergence of Newton's method (Equation (2)) given in Section 2. Some numerical examples are also given in Section 2, whereas Section 3 contains the work on Bratu's equation.

## 2. Semilocal Convergence

**Theorem 1** (Kantorovich's theorem [1]). *Let $E_1$ and $E_2$ be Banach spaces. Let also $F : \Omega \subseteq E_1 \to E_2$ be a twice continuously differentiable operator in the sense of Fréchet where $\Omega$ is a non-empty open and convex region. Assume:*

(i)　$x_0 \in \Omega$ *and there exists* $\Gamma_0 = [F'(x_0)]^{-1}$ *with* $\|\Gamma_0\| \leq \gamma$,

(ii)　$\|\Gamma_0 F(x_0)\| \leq \eta$,

(iii)　$\|F''(x)\| \leq k$, $\quad x \in \Omega$,

(iv)　$k\gamma\eta \leq \dfrac{1}{2}$,

(v)　$\overline{B(x_0, s^* - s_0)} \subseteq \Omega$, *where* $s^* = s_0 + \dfrac{1 - \sqrt{1 - 2k\gamma\eta}}{k\gamma}$.

*Then, Newton's sequence defined in Equation (2) converges to a solution $x_*$ of the equation $F(x) = 0$. Moreover, $x_n, x_* \in \overline{B(x_0, s^* - s_0)}$, for all $n = 0, 1, 2, \ldots$ Furthermore, the solution $x_*$ is unique in $B(x_0, s^{**} - s_0)$, where $s^{**} = s_0 + \dfrac{1 + \sqrt{1 - 2k\gamma\eta}}{k\gamma}$, if $k\gamma\eta < \dfrac{1}{2}$, and in $\overline{B(x_0, s^{**} - s_0)}$, if $k\gamma\eta = \dfrac{1}{2}$, for some $s_0 \geq 0$. Furthermore, the following error bounds hold*

$$\|x_{n+1} - x_n\| \leq |s_{n+1} - s_n|$$

*and*

$$\|x_{n*} - x_n\| \leq |s^* - s_n|,$$

*where*

$$\begin{cases} s_0 \text{ is given}, \\[2mm] s_{n+1} = s_n - \dfrac{f(s_n)}{f'(s_n)}, \quad n \geq 0, \end{cases}$$

*and*

$$f(t) = \frac{k}{2}(t - s_0)^2 - \frac{t - s_0}{\gamma} + \frac{\eta}{\gamma}.$$

The Kantorovich theorem can be improved as follows:

**Theorem 2.** *Let $E_1$ and $E_2$ be Banach spaces. Let $F : \Omega \subseteq E_1 \to E_2$ be a twice continuously differentiable operator in the sense of Fréchet. Assume:*

(i)　$x_0 \in \Omega$ *and there exists* $\Gamma_0 = [F'(x_0)]^{-1}$ *with* $\|\Gamma_0\| \leq \gamma$,

(ii)　$\|\Gamma_0 F(x_0)\| \leq \eta$,

(iii)　$\|F'(x) - F'(x_0)\| \leq k_0\|x - x_0\|$, $\quad x \in \Omega$,

(iv)　$\|F''(x)\| \leq \bar{k}$, $x \in \Omega_0 := \Omega \cap B\left(x_0, \dfrac{1}{\gamma k_0} + s_0\right)$,

(v)　$\tilde{k}\gamma\eta \leq \dfrac{1}{2}$, *where* $\tilde{k} = \max\{k_0, \bar{k}\}$,

(vi)　$\overline{B(x_0, \bar{s}^* - s_0)} \subseteq \Omega$, *where* $\bar{s}^* = s_0 + \dfrac{1 - \sqrt{1 - 2\tilde{k}\gamma\eta}}{\tilde{k}\gamma}$.

Then, sequence $\{x_n\}$ generated by Method (2) converges to $x_*$. Moreover, $x_n, x_* \in \overline{B(x_0, \bar{s}^* - s_0)}$, $n \geq 0$. Furthermore, the solution $x_*$ is unique in $B(x_0, \bar{s}^{**} - s_0)$, where

$$\bar{s}^{**} = s_0 + \frac{1 + \sqrt{1 - 2\tilde{k}\gamma\eta}}{\tilde{k}\gamma}, \quad \text{if} \quad \tilde{k}\gamma\eta < \frac{1}{2}$$

and in

$$\overline{B(x_0, \bar{s}^{**} - s_0)}, \quad \text{if} \quad \tilde{k}\gamma\eta = \frac{1}{2},$$

for some $s_0 \geq 0$. Furthermore, the following error bounds hold

$$\|x_{n+1} - x_n\| \leq |\bar{s}_{n+1} - \bar{s}_n|$$

and

$$\|x_{n*} - x_n\| \leq |\bar{s}^* - \bar{s}_n|,$$

where

$$\begin{cases} \bar{s}_0 \text{ is given,} \\ \bar{s}_{n+1} = \bar{s}_n - \dfrac{\overline{f}(\bar{s}_n)}{\overline{f}'(\bar{s}_n)}, \quad n \geq 0, \end{cases}$$

and

$$\overline{f}(t) = \frac{\tilde{k}}{2}(t - \bar{s}_0)^2 - \frac{t - \bar{s}_0}{\gamma} + \frac{\eta}{\gamma}.$$

**Proof.** The iterates $x_n$ stay in $\Omega_0$ by the proof of the Kantorovich theorem, which is a more precise location for the solution than $\Omega$, since $\Omega_0 \subseteq \Omega$. $\quad\square$

**Remark 1.** *If $k_0 = k = \bar{k}$, Theorem 1 reduces to the Kantorovich theorem, where k is the Lipschitz constant for $x \in \Omega$ used in [1]. We get $k_0 \leq k$ and $\bar{k} \leq k$ so $\tilde{k} \leq k$ holds in general.*

*Notice that*

$$k\gamma\eta \leq \frac{1}{2} \quad \text{implies} \quad \tilde{k}\gamma\eta \leq \frac{1}{2},$$

*so the Newton–Kantorovich sufficient convergence condition $k\gamma\eta \leq \frac{1}{2}$ has been improved and under the same effort, because the computation of k requires the computation of $k_0$ or $\bar{k}$ as special cases.*

*Moreover, notice that if $\Omega_1 = \Omega \cap B\left(x_0, \frac{1}{\gamma k_0} + s_0 - \|\Gamma_0 F(x_0)\|\right)$ provided that $k_0\gamma\|\Gamma_0 F(x_0)\| \leq 1$ and (iv) of Theorem 2 holds on $\Omega_1$ with $\bar{\bar{k}}$ replacing $\bar{k}$, then Theorem 2 can be extended even further with $\Omega_1$, $\tilde{\tilde{k}} = \max\{k_0, \bar{\bar{k}}\}$, replacing $\Omega_0$ and $\tilde{k}$, respectively, since $\Omega_1 \subseteq \Omega_0$, so $\tilde{\tilde{k}} \leq \tilde{k}$.*

Concerning majorizing sequences, define

$$\begin{cases} r_0 = 0, \quad r_1 = \eta, \\ r_{n+2} = r_{n+1} + \dfrac{\bar{k}(r_{n+1} - r_n)^2}{2(1 - k_0 r_{n+1})}, \end{cases}$$

$$\begin{cases} \bar{\bar{s}}_0 \quad \text{given,} \\ \bar{\bar{s}}_{n+1} = \bar{\bar{s}}_n - \dfrac{\overline{\overline{f}}(\bar{\bar{s}}_n)}{\overline{\overline{f}}'(\bar{\bar{s}}_n)}, \end{cases}$$

$$\begin{cases} \bar{r}_0 = 0, \quad \bar{r}_1 = \eta, \\ \bar{r}_{n+2} = \bar{r}_{n+1} + \dfrac{\bar{\bar{k}}(\bar{r}_{n+1} - \bar{r}_n)^2}{2(1 - k_0 \bar{r}_{n+1})}, \end{cases}$$

where

$$\bar{\bar{f}}(t) = \frac{\tilde{k}}{2}(t - \bar{\bar{s}}_0)^2 - \frac{t - \bar{\bar{s}}_0}{\gamma} + \frac{\eta}{\gamma}.$$

According to the proofs, $\{r_n\}$ and $\{\bar{r}_n\}$ are majorizing sequences tighter than $\{\bar{s}_n\}$ and $\{\bar{\bar{s}}_n\}$, respectively, and as such, they converge under the same convergence criteria. Notice also that $r^* = \lim\limits_{n \to +\infty} r_n \leq \bar{s}^*$ and $\bar{r}^* = \lim\limits_{n \to +\infty} \bar{r}_n \leq \bar{\bar{s}}^*$.

**Example 1.** *Let $F(x) = x^3 - p$, $p \in [0, \frac{1}{2}]$, $\Omega = B(x_0, 1 - p)$, $s_0 = \bar{s}_0 = \bar{\bar{s}}_0 = r_0 = \bar{r}_0 = 0$ and $x_0 = 1$.*

**Case 1.** *$\Omega := B(x_0, 1 - p)$. Then, we have that*

$$|\Gamma_0| = \frac{1}{3} = \gamma, \quad |\Gamma_0 F(x_0)| = \frac{1}{3}(1 - p) = \eta,$$

$$|F''(x)| \leq 6|x| \leq 6(|x_0 - x| + |x_0|) \leq 6(1 + 1 - p) = 6(2 - p) = k$$

*and*

$$|F'(x_0) - F'(x)| \leq 3|(x_0 + x)(x_0 - x)| \leq 3(|x_0 - x| + 2|x_0|)|x_0 - x| \leq k_0|x_0 - x|,$$

*so*

$$k_0 = 3(3 - p).$$

*We see that Kantorovich's result [4] (see Theorem 1) cannot be applied, since*

$$k\gamma\eta > \frac{1}{2} \quad \text{for all} \quad p \in \left[0, \frac{1}{2}\right).$$

**Case 2.** *$\Omega_0 := \Omega \cap B\left(x_0, \dfrac{1}{\gamma k_0}\right)$. Then, we get*

$$\begin{aligned} |F'(y) - F'(x)| &= 3|(y + x)(y - x)| \leq 3(|x_0 - x| + |x_0 - y| + 2|x_0|)|y - x| \\ &\leq 6\left[\frac{1}{\gamma k_0} + 1\right]|x - y| = \bar{k}|x - y| \end{aligned}$$

*where*

$$\bar{k} = 6\left(\frac{4 - p}{3 - p}\right),$$

*so by Theorem 2, Newton's method converges for*

$$\tilde{k} = \begin{cases} \bar{k}, & p \geq 2 - \sqrt{3}, \\ k_0, & 0 < p \leq 2 - \sqrt{3}, \end{cases}$$

*since*

$$\tilde{k}\gamma\eta = \frac{2}{3}\frac{(4 - p)(1 - p)}{3 - p} < \frac{1}{2} \quad \text{for all} \quad p \in \left[0.46198316\ldots, \frac{1}{2}\right).$$

**Case 3.** $\Omega_1 := \Omega \cap B\left(x_1, \dfrac{1}{\gamma k_0} - |\Gamma_0 F(x_0)|\right)$ *provided that* $k_0 \gamma \eta \leq 1$. *In this case, we obtain from*

$$
\begin{aligned}
|F'(x) - F'(y)| &\leq 3[|x - x_1| + |y - x_1| + 2|x_1|]|x - y| \\
&\leq 6\left[\left(\frac{1}{\gamma k_0} - \eta\right) + \frac{2+p}{3}\right]|x - y| \leq \overline{\overline{k}}|x - y|
\end{aligned}
$$

*so*

$$
\overline{\overline{k}} = -\frac{2(2p^2 - 5p - 6)}{3 - p}.
$$

*Therefore, we must have that*

$$
\tilde{\tilde{k}} \gamma \eta \leq \frac{1}{2}
$$

*and*

$$
k_0 \gamma \eta < 1
$$

*which are true for* $p \in \left[0.42973177\ldots, \dfrac{1}{2}\right)$ *since* $k_0 < \overline{\overline{k}}$, *so* $\tilde{\tilde{k}} = \overline{\overline{k}}$. *Hence, we have extended the convergence interval of the previous cases.*

The sufficient convergence criterion for the modified Newton's method

$$
\begin{cases}
x_0 \text{ given in } \Omega, \\
x_{n+1} = x_n - [F'(x_0)]^{-1} F(x_n), \quad n \geq 0
\end{cases}
$$

is the same as the Kantorovich condition $(iv)$. In [7], though we proved that this condition can be replaced by $k_0 \gamma \eta \leq \frac{1}{2}$ which is weaker if $k_0 < k$. In the case of the example at hand, we have that this condition is satisfied as in the previous case interval. Therefore, by restricting the convergence domain, sufficient convergence criteria can be obtained for Newton's method identical to the ones required for the convergence of the modified Newton's method. The same advantages are obtained if the preceding Lipschitz constants are replaced by the $\psi$ functions that follow.

It is worth noting that the center-Lipschitz condition (not introduced in earlier studies) makes it possible to restrict the domain from $\Omega$ to $\Omega_0$ (or $\Omega_1$), where the iterates actually lie and where

$$
\|F'(x)^{-1}\| \leq \frac{\gamma}{1 - \gamma k_0 \|x - x_0\|}
$$

can be used instead of the less tight estimate

$$
\|F'(x)^{-1}\| \leq \frac{\gamma}{1 - \gamma k \|x - x_0\|}
$$

used in Theorem 1 and in other related earlier studies using only condition $(iv)$ in Theorem 1.

Next, the condition

$$
\|F''(\overline{x})\| \leq k, \quad \overline{x} \in \Omega
$$

is replaced by

$$
\|F''(\overline{x})\| \leq \psi(\|\overline{x}\|), \quad \overline{x} \in \Omega. \tag{5}
$$

Next, we show how to improve these results by relaxing Equation (5) using even weaker conditions

$$
\|F'(\overline{x}) - F'(\overline{x}_0)\| \leq v(\|\overline{x} - \overline{x}_0\|), \quad \overline{x} \in \Omega, \tag{6}
$$

where $v : [t_0, +\infty) \cup \{0\} \to \mathbb{R}$ is a non-decreasing continuous function satisfying $v(t_0) \geq 0$. Suppose that equation $\gamma v(t - t_0) = 1$ has at least one positive solution. Denote by $\rho_1$ the smallest such solution.

Moreover, suppose that

$$\|F''(\overline{x})\| \le \psi_1(\|\overline{x}\|), \quad \overline{x} \in \Omega_0 = \Omega \cap B(\overline{x}_0, \rho_1 - t_0) \tag{7}$$

or Equation (6) and

$$\|F''(\overline{x})\| \le \psi_2(\|\overline{x}\|), \quad \overline{x} \in \Omega_1 = \Omega \cap B(\overline{x}_1, \rho_1 - t_0 - \|\Gamma_0 F(\overline{x}_0)\|), \quad \text{if} \quad \|\Gamma_0 F(\overline{x}_0)\| \le \rho_1 - t_0, \tag{8}$$

where $\psi_1, \psi_2 : [\rho_1 - t_0, +\infty) \cup \{0\} \to \mathbb{R}$ are non-decreasing functions.

If function $v$ is strictly increasing, then we can choose $\rho_1 = v^{-1}\left(\frac{1}{\gamma}\right) + t_0$.

Notice that Equation (5) implies Equations (6) and (7) or Equations (6) and (8) but not necessarily vice versa. Moreover, we have that

$$v(t) \le \psi(t), \tag{9}$$

$$\psi_1(t) \le \psi(t) \tag{10}$$

and

$$\psi_2(t) \le \psi(t). \tag{11}$$

Next, we show that $\psi_1$ or $\psi_2$ can replace $\psi$ in the results obtained in Reference [4]. Then, in view of Equations (9)–(11), the new results are finer and are provided without additional cost, since $\psi$ requires the computation functions $v$, $\psi_1$ and $\psi_2$ as special cases. Notice that function $v$ is needed to determine $\rho_1$ (i. e., $\Omega_0$ and $\Omega_1$) and that $\Omega_0 \subseteq \Omega$ and $\Omega_1 \subseteq \Omega_0$.

## 3. Bratu's Equation

Bratu's equation is defined by the following nonlinear integral equation

$$x(t_1) = \mu \int_\alpha^\beta T(t_1, t_2) e^{x(t_2)} \, dt_2, \quad t_1 \in [\alpha, \beta], \tag{12}$$

where $-\infty < \alpha < \beta < \infty$, $\mu \in \mathbb{R}_+$ and the kernel $T$ is the Green's function

$$T(t_1, t_2) = \begin{cases} \dfrac{(\beta - t_1)(t_2 - \alpha)}{\beta - \alpha}, & t_2 \le t_1, \\ \dfrac{(t_1 - \alpha)(\beta - t_2)}{\beta - \alpha}, & t_1 \le t_2. \end{cases}$$

Observe that Equation (12) can also be seen as the following boundary value problem [8]:

$$\begin{cases} \dfrac{d^2 x(t_2)}{dt_2^2} + \mu \, e^{x(t_2)} = 0, \\ x(\alpha) = x(\beta) = 0. \end{cases}$$

Let $\mu > 0$ and $\alpha = 0$. It follows from [8] that Equation (12) has two solutions such that $x_1(t_2) \ne x_2(t_2)$, provided that for for each $\mu \in (0, \mu_1)$, where $\mu_1 = \frac{3.51375\ldots}{\beta^2}$. Next, we show a sketch of both solutions in Figure 1.

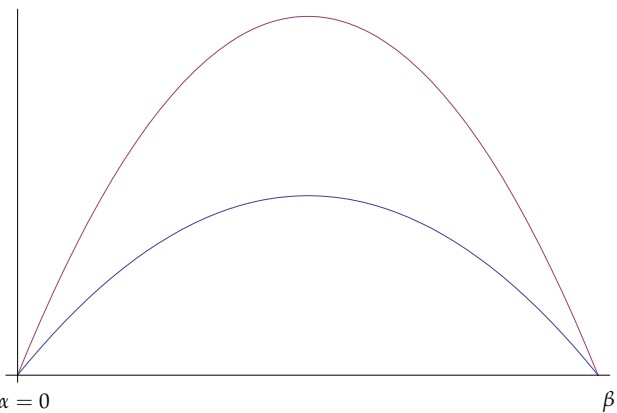

**Figure 1.** The two real solutions of Bratu's Equation (12).

Bratu's equation appears in connection to many problems: combustion, heat transfer, chemical reactions, and nanotechnology [9].

Using Newton's method, we approximate the solutions of Bratu's equation. Let $F : \Omega \subseteq \mathcal{C}([\alpha, \beta]) \to \mathcal{C}([\alpha, \beta])$ be defined by

$$[F(x)](t_1) = x(t_1) - \mu \int_\alpha^\beta T(t_1, t_2) e^{x(t_2)} \, dt_2. \tag{13}$$

But condition (3) does not hold if operator (13) is defined by Equation (13), since

$$[F'(x)y](t_1) = y(t_1) - \mu \int_\alpha^\beta T(t_1, t_2) e^{x(t_2)} y(t_2) \, dt_2,$$

$$[F''(x)yz](t_1) = -\mu \int_\alpha^\beta T(t_1, t_2) e^{x(t_2)} z(t_2) y(t_2) \, dt_2,$$

Therefore, it is clear that $\|F''(x)\|$ is not bounded in a general domain $\Omega$. However, it is hard to find a region containing a solution of $F(x) = 0$ and such that $\|F''(x)\|$ is bounded there.

Our aim is to solve $F(\overline{x}) = 0$ using Newton's method

$$\begin{cases} \overline{x}_0 \text{ given in } \Omega, \\ \overline{x}_{n+1} = \overline{x}_n - [F'(\overline{x}_n)]^{-1} F(\overline{x}_n) \quad \text{for all} \quad n = 0, 1, 2, \ldots \end{cases} \tag{14}$$

Then, we solve

$$F'(\overline{x}_n)(\overline{x}_{n+1} - \overline{x}_n) = -F(\overline{x}_n). \tag{15}$$

Using $m$ nodes in the Gauss-Legendre quadrature formula

$$\int_\alpha^\beta f(t) \, dt \simeq \sum_{i=1}^m \beta_i f(t_i),$$

where the nodes $t_i$ and the weights $\beta_i$ are known. We can write

$$x_i = \mu \sum_{j=1}^m a_{ij} e^{x_j}, \quad i = 1, 2, \ldots, m, \quad \text{where} \quad a_{ij} = \begin{cases} \beta_j \dfrac{(\beta - t_i)(t_j - \alpha)}{\beta - \alpha} & \text{if } j \leq i, \\ \beta_j \dfrac{(\beta - t_j)(t_i - \alpha)}{\beta - \alpha} & \text{if } j > i, \end{cases}$$

or

$$F(\overline{x}) \equiv \overline{x} - \mu A u(\overline{x}) = 0, \tag{16}$$

where

$$\overline{x} = (x_1, \ldots, x_m)^T, \quad A = (a_{ij})_{i,j=1}^m \quad \text{and} \quad u(\overline{x}) = (e^{x_1}, \ldots, e^{x_m})^T.$$

We shall relate sequence $\{\overline{x}_n\}$ with its majorizing sequence

$$\begin{cases} s_0 \text{ given,} \\ s_{n+1} = s_n - \dfrac{f(s_n)}{f'(s_n)} \quad \text{for all} \quad n = 0, 1, 2, \ldots. \end{cases}$$

To achieve this using Equation (16), we compute $F'$, $F''$ and

$$F'(\overline{x})\overline{y} = (I - \mu A D(\overline{x}))\overline{y}, \quad D(\overline{x}) = \mathrm{diag}\{e^{x_1}, e^{x_2}, \ldots, e^{x_n}\},$$

where $\overline{y} \in \mathbb{R}^m$,

$$F''(\overline{x})\overline{y}\,\overline{z} = -\mu A (e^{x_1}y_1 z_1, e^{x_2}y_2 z_2, \ldots, e^{x_m}y_m z_m)^T,$$

$\overline{y} = (y_1, y_2, \ldots, y_m)$, and $\overline{z} = (z_1, z_2, \ldots, z_m)$. Let $B(x, \rho) = \{y \in \mathbb{R}^m; \|y - x\| \le \rho\}$ and let $\overline{B(x, \rho)}$ be its closure.

Clearly, Theorems 1 and 2 hold if operator $F$ is defined by Equation (16) and Newton's method in the form of Equation (14) is used.

We shall verify the hypotheses of these theorems, so we can solve our problem. To achieve this, $\mu\|A\|\|D(\overline{x}_0)\| < 1$ sets

$$\|\Gamma_0\| = \frac{1}{1 - \mu\|A\|\|D(\overline{x}_0)\|} = \gamma, \tag{17}$$

and

$$\|\Gamma_0 F(\overline{x}_0)\| = \frac{\|\overline{x}_0 - \mu A u(\overline{x}_0)\|}{1 - \mu\|A\|\|D(\overline{x}_0)\|} = \eta, \tag{18}$$

where $u(\overline{x}_0) = (e^{\widehat{x}_1}, e^{\widehat{x}_2}, \ldots, e^{\widehat{x}_m})^T$ and $\overline{x}_0 = (\widehat{x}_1, \widehat{x}_2, \ldots, \widehat{x}_m)^T$. Moreover, we have

$$\|F''(\overline{x})\overline{y}\,\overline{z}\| \le \mu\|A\| \left\| (e^{x_1}y_1 z_1, e^{x_2}y_2 z_2, \ldots, e^{x_m}y_m z_m)^T \right\|$$

and $\|F''(\overline{x})\|_\infty \le \mu\|A\|_\infty e^{\|\overline{x}\|_\infty}$, where we used the infinity norm. Notice that $\|F''(\overline{x})\|_\infty$ is not bounded, since $e^{\|\overline{x}\|_\infty}$ is increasing as a function of $\|\overline{x}\|_\infty$. Hence, Theorem 1 or Theorem 2 cannot be used.

**Remark 2.** *Notice that Kantorovich's Theorem 1 cannot apply, although $F'$ is Lipschitz continuous.*

We look for a bound for $\|F''(x)\|_\infty$ in such domain ([6]). If $\overline{x}_*$ solves Equation (16), we have $\|\overline{x}_*\|_\infty \in [0, r_1] \cup [r_2, +\infty)$, where $r_1$ and $r_2$ ($0 < r_1 < r_2$) are roots of the scalar equation $t - \mu\|A\|_\infty e^t = 0$. (See Figure 2). We choose $\overline{x}_0$ such that $\overline{x}_0 \in B(0, \rho)$ with $\rho \in (r_1, r_2)$.

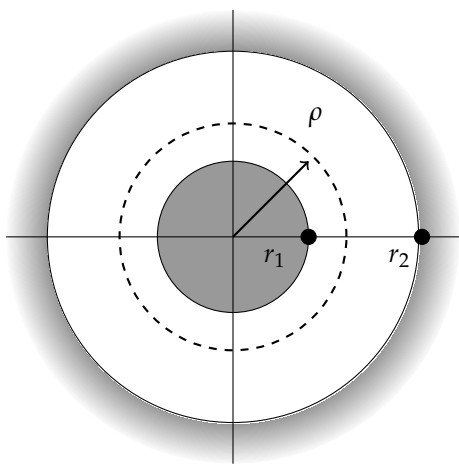

**Figure 2.** $\|\overline{x}_*\|_\infty \in [0, r_1] \cup [r_2, +\infty)$.

**Example 2.** *Let us consider Bratu's Equation (12) with $\mu = 1$, $\alpha = 0$ and $\beta = 1$ to obtain $r_1 = 0.14247951\ldots$ and $r_2 = 3.27838858\ldots$ By choosing $\rho = 3$, $m = 8$, $\overline{x}_0 = \overline{0} = (0, 0, \ldots, 0)^T$, we see that with $s_0 = 0$*

$$\|F''(\overline{x})\|_\infty \le (0.12355899\ldots)\, e^3 = 2.48174869\ldots = k,$$

$$\overline{\overline{k}} = 0.31123266\ldots, \quad \overline{k} = 0.35736407\ldots, \quad k_0 = 0.82724956\ldots$$

$$\overline{\overline{k}} < \overline{k} < k_0 < k$$

$$\gamma = 1.13821435\ldots, \quad \eta = 0.13821435\ldots$$

*so $\tilde{k} = \tilde{\tilde{k}} = k_0$,*

$$k\gamma\eta = 0.39042265\ldots < \frac{1}{2}, \quad k_0\gamma\eta = 0.13014088\ldots < \frac{1}{2},$$

$$B(\overline{x}_0, s^*) \subseteq \Omega = B(\overline{0}, \rho), \quad where \quad s^* = 0.18828503\ldots$$

$$B(\overline{x}_0, \overline{s}^*) \subseteq \Omega_0, \quad where \quad \overline{s}^* = 0.14861209\ldots$$

$$B(\overline{x}_0, \overline{\overline{s}}^*) \subseteq \Omega_1, \quad where \quad \overline{\overline{s}}^* = 0.14861209\ldots$$

*The conditions of Theorem 2 hold.*

*Consequently, we obtain the solution $\overline{x}_* = (x_{*1}, \ldots, x_{*8})^T$ after three iterations (see Table 1).*

**Table 1.** The solution $\overline{x}_*$ of Equation (12) for $\mu = 1$, $\alpha = 0$ and $\beta = 1$.

| $n$ | $x_{*n}$ |
| --- | --- |
| 1 | $0.0109342501\ldots$ |
| 2 | $0.0518018395\ldots$ |
| 3 | $0.1036467688\ldots$ |
| 4 | $0.1393014227\ldots$ |
| 5 | $0.1393014227\ldots$ |
| 6 | $0.1036467688\ldots$ |
| 7 | $0.0518018395\ldots$ |
| 8 | $0.0109342501\ldots$ |

*Concerning Theorem 1, we define*

$$f(t) = \frac{k}{2}(t - s_0)^2 - \frac{t - s_0}{\gamma} + \frac{\eta}{\gamma}, \quad with \quad \gamma = -\frac{1}{f'(s_0)} \quad and \quad \eta = -\frac{f(s_0)}{f'(s_0)}, \tag{19}$$

*as an auxiliary function to construct majorazing sequence $\{s_n\}$. We also use the sequence*

$$
\begin{cases}
s_0 = 0, \\
s_{n+1} = s_n - \dfrac{f(s_n)}{f'(s_n)}, & n \geq 0.
\end{cases}
\tag{20}
$$

*Note then that* $\lim\limits_{n \to +\infty} s_n = s^* = \dfrac{1 - \sqrt{1 - 2k\gamma\eta}}{k\gamma}$, $\lim\limits_{n \to +\infty} \bar{s}_n = \bar{s}^* = \dfrac{1 - \sqrt{1 - 2\tilde{k}\gamma\eta}}{\tilde{k}\gamma}$,
$\lim\limits_{n \to +\infty} r_n = r^* = 0.1420714278\ldots$ *and* $\lim\limits_{n \to +\infty} \bar{r}_n = \bar{r}^* = 0.1415728924\ldots$. *We also obtain the a priori error estimates shown in Table 2, which shows that the error bounds are improved under our new approach.*

**Table 2.** Absolute error and a priori error estimates.

| $n$ | $\|\bar{x}_* - \bar{x}_n\|$ | $|\bar{r}^* - \bar{r}_n|$ | $|r^* - r_n|$ | $|\bar{\bar{s}}^* - \bar{\bar{s}}_n|$ | $|\bar{s}^* - \bar{s}_n|$ | $|s^* - s_n|$ |
|---|---|---|---|---|---|---|
| 0 | $0.1393014227\ldots$ | $0.1415728924\ldots$ | $0.1420714278\ldots$ | $0.1486120940\ldots$ | $0.1486120940\ldots$ | $0.1882850304\ldots$ |
| 1 | $0.0010870679\ldots$ | $0.0033585375\ldots$ | $0.0038570729\ldots$ | $0.0103977391\ldots$ | $0.0103977391\ldots$ | $0.0500706756\ldots$ |
| 2 | $0.0000000702\ldots$ | $0.0000019858\ldots$ | $0.0000030075\ldots$ | $0.0000585139\ldots$ | $0.0000585139\ldots$ | $0.0058088471\ldots$ |
| 3 | $2.775557 \times 10^{-17}$ | $1.389915 \times 10^{-12}$ | $1.831368 \times 10^{-12}$ | $1.874082 \times 10^{-9}$ | $1.874082 \times 10^{-9}$ | $0.0000983546\ldots$ |

In this section, we consider the alternative to Equation (4) condition

$$
\|F''(\bar{x})\| \leq \psi_1(\|\bar{x}\|) \leq \psi(t - t_0 + \|\bar{x}_0\|) \quad \text{for} \quad \|\bar{x} - \bar{x}_0\| \leq t - t_0.
$$

since $\psi_1$ is non-decreasing. Then, we look for a function $f_1$

$$
-\frac{1}{f_1'(t_0)} = \gamma, \qquad -\frac{f_1(t_0)}{f_1'(t_0)} = \eta \qquad \text{and} \qquad f_1''(t) = \psi_1(t - t_0 + \|\bar{x}_0\|).
\tag{21}
$$

The solution of Equation (21) is given by

$$
f_1(t) = \int_{t_0}^{t} \int_{t_0}^{\theta} \psi_1(\xi - t_0 + \|\bar{x}_0\|) \, d\xi \, d\theta - \frac{t - t_0}{\gamma} + \frac{\eta}{\gamma}.
\tag{22}
$$

Define also

$$
f_0(t) = \int_{t_0}^{t} \int_{t_0}^{\theta} v(\xi - t_0 + \|\bar{x}_0\|) \, d\xi \, d\theta - \frac{t - t_0}{\gamma} + \frac{\eta}{\gamma}.
$$

We suppose in what follows that

$$
f_0(t) \leq f(t).
$$

Otherwise, i.e., if $f(t) \leq f_0(t)$, then the following results hold with $f_0$ replacing $f_1$.
Notice that $f_1$ is the function obtained by Kantorovich if $t_0 = s_0$ and $\psi_1(t - t_0 + \|\bar{x}_0\|) = \tilde{k}$, $\tilde{k} \in \mathbb{R}$.
For Bratu's equation, we have $\psi_1(t - t_0 + \|\bar{x}_0\|) = \mu\|A\|e^{t - t_0 + \|\bar{x}_0\|}$ and function (22) is reduced to

$$
f_1(t) = \mu\|A\|e^{\|\bar{x}_0\| - t_0} \left( e^t + e^{t_0}(t_0 - 1 - t) \right) - \frac{t - t_0}{\gamma} + \frac{\eta}{\gamma},
\tag{23}
$$

with $\gamma$ and $\eta$ defined in Equations (17) and (18), respectively. Next, we need the auxiliary results for function $f_1$.

**Lemma 1.** *Let $f_1$ be the function defined in Equation (23) and*

$$
\alpha_1 = \ln \left( e^{t_0} + \frac{e^{t_0 - \|\bar{x}_0\|}}{\gamma\mu\|A\|} \right).
\tag{24}
$$

*Then:*

(*a*)  $\alpha_1$ *is the unique minimum of $f_1$ in $[t_0, +\infty)$.*

(*b*)  $f_1$ *is non-increasing in $(t_0, \alpha_1)$.*

(*c*)  *If $\alpha_1 \geq \dfrac{1 + t_0 + \eta + t_0 \gamma \mu \|A\| e^{\|\overline{x}_0\|}}{1 + \gamma \mu \|A\| e^{\|\overline{x}_0\|}}$, the equation $f_1(t) = 0$ has at least one root in $(t_0, +\infty)$. If $t^*$ is the smallest positive root of $f_1(t) = 0$, we have $t_0 < t^* \leq \alpha_1$.*

Next, we define the scalar sequence

$$t_0 \text{ given,} \quad t_{n+1} = t_n - \frac{f_1(t_n)}{f_1'(t_n)} \quad \text{for all} \quad n = 0, 1, 2 \ldots \tag{25}$$

**Lemma 2.** *If*

$$\alpha_1 \geq \frac{1 + t_0 + \eta + t_0 \gamma \mu \|A\| e^{\|\overline{x}_0\|}}{1 + \gamma \mu \|A\| e^{\|\overline{x}_0\|}}, \tag{26}$$

*where $f_1$, $\alpha_1$ are given in Equations (23) and (24), respectively, then sequence (25) is increasingly convergent to the smallest positive root $t^*$ of $f_1(t) = 0$.*

We need an auxiliary result relating sequence $\{\overline{x}_n\}$ to $\{t_n\}$.

**Lemma 3.** *Let $F : \Omega \subseteq \mathbb{R}^m \to \mathbb{R}^m$. Let $f_1$ be the function defined in Equation (23) and $\alpha_1$ in Equation (24). If condition (26) is satisfied, then $\overline{x}_n \in B(\overline{x}_0, t^* - t_0)$, for $n \geq 1$, where $t^*$ is the smallest positive root of $f_1(t) = 0$. Then, sequence (25) is majorizing for the sequence $\{\overline{x}_n\}$:*

$$\|\overline{x}_{n+1} - \overline{x}_n\| \leq t_{n+1} - t_n, \quad n \geq 0.$$

**Proof.**  Observe that

$$\|\overline{x}_1 - \overline{x}_0\| \leq \eta = t_1 - t_0 < t^* - t_0.$$

We prove the following four items for all $n \geq 1$:

(*i*)   *There exists $\Gamma_n = [F'(\overline{x}_n)]^{-1}$ such that $\|\Gamma_n\| \leq -f_1'(t_n)^{-1}$,*

(*ii*)   $\|F(\overline{x}_n)\| \leq f_1(t_n)$,

(*iii*)   $\|\overline{x}_{n+1} - \overline{x}_n\| \leq t_{n+1} - t_n$,

(*iv*)   $\|\overline{x}_{n+1} - \overline{x}_0\| \leq t^* - t_0$.

Firstly, from

$$\begin{aligned}
\|I - \Gamma_0 F'(\overline{x}_1)\| &\leq \|\Gamma_0\| \int_0^1 \left\| F''\left(\overline{x}_0 + \theta(\overline{x}_1 - \overline{x}_0)\right) \right\| d\theta \|\overline{x}_1 - \overline{x}_0\| \\
&\leq \gamma \mu \|A\| e^{\|\overline{x}_0\|} (e^{t_1 - t_0} - 1) \\
&\leq 1 - \frac{f_1'(t_1)}{f_1'(t_0)} < 1,
\end{aligned}$$

$\Gamma_1$ exists and

$$\|\Gamma_1\| \leq \frac{\|\Gamma_0\|}{1 - \|I - \Gamma_0 F'(\overline{x}_1)\|} \leq -f_1'(t_1)^{-1}.$$

Secondly, from Taylor's series and Equation (14),

$$F(\overline{x}_1) = \int_0^1 F''\left(\overline{x}_0 + \theta(\overline{x}_1 - \overline{x}_0)\right)(1 + \theta) \, d\theta (\overline{x}_1 - \overline{x}_0)^2,$$

it follows that

$$
\begin{aligned}
\|F(\bar{x}_1)\| &\leq \int_0^1 \mu\|A\|e^{t_0+\theta(t_1-t_0)-t_0+\|\bar{x}_0\|}(1+\theta)\,d\theta\,(t_1-t_0)^2 \\
&= \mu\|A\|e^{\|\bar{x}_0\|-t_0}\left(e^{t_1}-(1+t_1-t_0)e^{t_0}\right) \\
&= f_1(t_1),
\end{aligned}
$$

since $\|\bar{x}_0+\theta(\bar{x}_1-\bar{x}_0)-\bar{x}_0\| \leq \theta\|\bar{x}_1-\bar{x}_0\| \leq \theta(t_1-t_0) = t_0+\theta(t_1-t_0)-t_0$.

Thirdly,

$$
\|\bar{x}_2-\bar{x}_1\| = \|\Gamma_1 F(\bar{x}_1)\| \leq -\frac{f_1(t_1)}{f_1'(t_1)} = t_2-t_1.
$$

Fourthly,

$$
\|\bar{x}_2-\bar{x}_0\| \leq \|\bar{x}_2-\bar{x}_1\| + \|\bar{x}_1-\bar{x}_0\| \leq t_2-t_1+t_1-t_0 = t_2-t_0 \leq t^*-t_0.
$$

Then, if $(i)$–$(iv)$ hold for all $n = 0,1,2,\ldots,k$, we show in an analogous way that these items hold for $n = k+1$ too. $\quad\square$

The $(C)$ conditions shall be used:

$(C_1)$ $\bar{x}_0 \in \Omega$ and there exists $\Gamma_0 = [F'(\bar{x}_0)]^{-1}$ such that $\|\Gamma_0\| \leq \gamma$,
$(C_2)$ $\|\Gamma_0 F(\bar{x}_0)\| \leq \eta$,
$(C_3)$ $\|F''(\bar{x})\| \leq \psi_1(t-t_0+\|\bar{x}_0\|)$ for $\|\bar{x}-\bar{x}_0\| \leq t-t_0$,
$(C_4)$ $\overline{B(\bar{x}_0,t^*-t_0)} \subseteq \Omega$, where $t^*$ is the smallest root of the equation $f_1(t) = 0$ in $[t_0,+\infty)$.

Notice that $f_1'$ is increasing and $f_1'(t) > 0$ in $(\alpha_1,+\infty)$, since $\psi_1(0) > 0$, so that $f_1$ is strictly increasing in $(\alpha_1,+\infty)$. Hence, $f_1(t^*) = f_2(t^{**})$ with $t^* \leq t^{**}$.

**Theorem 3.** *Assume conditions $(C_1)$–$(C_4)$ are satisfied. If condition (26) is also satisfied, Newton's sequence given by Equation (14) converges to a solution $\bar{x}_*$ of Equation (16). Moreover, $\bar{x}_n, \bar{x}_* \in \overline{B(\bar{x}_0,t^*-t_0)}$ and $\|\bar{x}_*-\bar{x}_n\| \leq t^*-t_n$, for all $n \geq 0$, where $\{t_n\}$ is defined in Equation (25). Furthermore, if $t^{**} > t^*$, the solution $\bar{x}_*$ is unique in $B(\bar{x}_0,t^{**}-t_0) \cap \Omega$.*

**Proof.** Sequence $\{x_n\}$ converges, since $\{t_n\}$ is its majorizing sequence. Then, if $\bar{x}_* = \lim_{n\to+\infty}\bar{x}_n$, $\|\bar{x}_*-\bar{x}_n\| \leq t^*-t_n$, for all $n \geq 0$. Moreover, the sequence $\{\|F'(\bar{x}_n)\|\}$ is bounded. By the continuity of $F$, we have $F(\bar{x}_*) = 0$, since $\|F(\bar{x}_n)\| = \|F'(\bar{x}_n)(\bar{x}_{n+1}-\bar{x}_n)\| \leq \|F'(\bar{x}_n)\|\|\bar{x}_{n+1}-\bar{x}_n\|$ and $\lim_{n\to+\infty}\|\bar{x}_{n+1}-\bar{x}_n\| = 0$.

To show the uniqueness of $\bar{x}_*$, let $\bar{y}_*$ be another solution of Equation (16) in $B(\bar{x}_0,t^{**}-t_0) \cap \Omega$. Notice that $\bar{y}_* = \bar{x}_*$. From

$$
0 = F(\bar{y}_*) - F(\bar{x}_*) = \int_{\bar{x}_*}^{\bar{y}_*} F'(x)\,dx = \int_0^1 F'(\bar{x}_* + \theta(\bar{y}_*-\bar{x}_*))\,d\theta\,(\bar{y}_*-\bar{x}_*),
$$

it follows that $\bar{x}_* = \bar{y}_*$, provided that the operator $Q = \int_0^1 F'(\bar{x}_* + t(\bar{y}_*-\bar{x}_*))\,dt$ is invertible. To prove that $Q$ is invertible, we prove equivalently that there exists the operator $P^{-1}$, where $P = \Gamma_0 \int_0^1 F'(\bar{x}_* + \theta(\bar{y}_*-\bar{x}_*))\,d\theta$. Indeed, as

$$
\left\| I - \Gamma_0 \int_0^1 F'(\bar{x}_* + \theta(\bar{y}_*-\bar{x}_*))\,d\theta \right\| \leq \|\Gamma_0\| \left\| \int_0^1 \int_{\bar{x}_0}^{\bar{x}_*+\theta(\bar{y}_*-\bar{x}_*)} F''(z)\,dz\,d\theta \right\|
$$

$$
\leq \gamma \int_0^1 \int_0^1 \|F''(\bar{x}_0 + s((\bar{x}_*-\bar{x}_0) + \theta(\bar{y}_*-\bar{x}_*)))\| \, \|\bar{x}_*-\bar{x}_0 + \theta(\bar{y}_*-\bar{x}_*)\| \, ds\,d\theta
$$

$$< \gamma \int_0^1 \left( ((1-\theta)\|\overline{x}_* - \overline{x}_0\| + \theta\|\overline{y}_* - \overline{x}_0\|) \int_0^1 \mu\|A\| e^{\|\overline{x}_0\| + s(t^*-t_0+\theta(t^{**}-t^*))} \, ds \right) d\theta$$

$$= \gamma\mu\|A\| e^{\|\overline{x}_0\|} \left( \frac{e^{t^{**}-t_0} - e^{t^*-t_0}}{t^{**} - t^*} - 1 \right) = 1,$$

so $P^{-1}$ exists. □

**Remark 3.** *We have by Equation (22) that $f_1(t + t_0) = g(t)$, where*

$$g(t) = \int_0^t \int_0^\theta \psi_1(\xi + \|\overline{x}_0\|) \, d\xi \, d\theta - \frac{t}{\gamma} + \frac{\eta}{\gamma}.$$

**Remark 4.**

(a) *If $v = \psi = \psi_1$, the results in this study coincide with the ones in [4]. Moreover, if inequality in Equations (9)–(11) is strict, then, the new results have the following advantages: weaker sufficient convergence conditions, tighter error estimates on $\|\overline{x}_{n+1} - \overline{x}_n\|$, $\|\overline{x}_n - \overline{x}_*\|$ and at least as precise information on the location of the solution $\overline{x}_*$.*

(b) *These results can be improved even further, if we simply use the condition*

$$\|F''(\overline{x})\| \le \psi_2(t - t_1 + \|\overline{x}_1\|), \quad \|\overline{x} - \overline{x}_1\| \le t - t_1,$$

*and majorizing function $f_2$ (as in $f_1$ with $\psi_1 = \psi_2$, $t_0 = t_1$) (also see the numerical section).*

**Remark 5.**

(a) *It is worth noting that there are alternative approaches to the root-finding other than Newton's method [10,11], where the latter one has cubic order of convergence, whereas Newton's is only quadratic.*

(b) *If the solution is sufficiently smooth, then one can use generalized Gauss quadrature rules for splines. This way, instead of projecting $f$ into a space of higher-degree polynomials as is done in our article, one can project it to a spline space (see [12–14]). These quadratures in general do not affect the convergence order, but they do make the computation more efficient, since fewer quadrature points are required to reach a certain error tolerance.*

## 4. Specialized Bratu's Equation

Consider the equation

$$x(t_1) = \int_0^1 T(t_1, t_2) \, e^{x(t_2)} \, dt_2, \quad t_1 \in [0,1]. \tag{27}$$

We transform Equation (27) into a finite dimensional problem, as we have done above, with $m = 8$, so that Equation (27) is equivalent to Equation (16) with $\mu = 1$, $\alpha = 0$, $\beta = 1$. For this case, we have

$$F'(\overline{x})\overline{y} = (I_8 - A\operatorname{diag}\{e^{x_1}, e^{x_2}, \ldots, e^{x_8}\})\overline{y}$$

where $\overline{y} \in \mathbb{R}^m$, and

$$F''(\overline{x})\overline{y}\,\overline{z} = -A(e^{x_1}y_1z_1, e^{x_2}y_2z_2, \ldots, e^{x_8}y_8z_8)^T,$$

where $\overline{y} = (y_1, y_2, \ldots, y_m)$ and $\overline{z} = (z_1, z_2, \ldots, z_m)$.

In Section 2, we have seen that $\|F''(\overline{x})\|_\infty \le \|A\|_\infty e^{\|\overline{x}\|_\infty}$, so that $\|F''(\overline{x})\|_\infty$ is not bounded. Then, any solution $\overline{x}_*$ of the particular system given by Equation (16) should satisfy $\|\overline{x}_*\|_\infty \le \|A\|_\infty e^{\|\overline{x}_*\|_\infty}$. We can take the region $B(0, \rho)$, with $\rho \in (r_1, r_2)$ and $r_1 = 0.14247951\ldots$ and $r_2 = 3.27838858\ldots$, where $\|F''(\overline{x})\|_\infty$ is bounded and contains the solution $\overline{x}_*$ (see Figure 2). The convergence of Newton's method to $\overline{x}_*$ follows Kantorovich's Theorem 1.

In Theorem 3, set $t_0 = 0$ and $\rho = 3$ (according to Remark 3), we have

$$\|F''(\overline{x})\|_\infty \le \|A\| \, e^{t + \|\overline{x}_0\|_\infty} \le \|A\| e^3,$$

so, we can choose $\psi(t) = \|A\| e^3$ and

$$f(t) = \|A\| \int_0^3 \int_0^\theta e^3 \, d\xi \, d\theta - \frac{t}{\gamma} + \frac{\eta}{\gamma}.$$

Then, function $f(t)$ is defined by

$$f(t) = \frac{9}{2} \|A\| e^3 - \frac{t}{\gamma} + \frac{\eta}{\gamma}.$$

Using conditions (6) and (7), we have

$$v(t) = \mu \|A\| e^t,$$

$$\rho_1 = 1.961575\ldots$$

and

$$\psi_1(t) = \|A\| e^{\rho_1}.$$

Then, we define

$$f_1(t) = \|A\| \int_0^{\rho_1} \int_0^\theta e^{\rho_1} \, d\xi \, d\theta - \frac{t}{\gamma} + \frac{\eta}{\gamma}$$

so

$$f_1(t) = \frac{\rho_1^2}{2} \|A\| e^{\rho_1} - \frac{t}{\gamma} + \frac{\eta}{\gamma}.$$

Next, we find the solutions $t^*$ and $t_1^*$ of the equations $f(t) = 0$ and $f_1(t) = 0$ to be, respectively:

$$t^* = \frac{9}{2} \gamma \|A\| e^3 + \eta = 12.849643\ldots$$

and

$$t_1^* = \frac{9}{2} \gamma \|A\| e^{\rho_1} + \eta = 2.062104\ldots.$$

We see that $t_1^* \in (r_1, r_2)$ but $t^* \notin (r_1, r_2)$. Then, the results in [4] cannot assure convergence to $\overline{x}_*$ but our results guarantee convergence.

Moreover, we have that

$$t_0 < t^* = 2.062104\ldots < \alpha_1 = 2.0931624\ldots.$$

## 5. Conclusions

In this article, we first introduce new Kantorovich-type results for the semilocal convergence on Newton's method for Banach space valued operators using our idea of convergence regions. Hence, we expand the applicability of Newton's method. Then, we focus our results on solving Bratu's equation.

**Author Contributions:** These authors contributed equally to this work.

**Funding:** This research received no external funding.

**Acknowledgments:** We would like to express our gratitude to the anonymous reviewers for their help with the publication of this paper.

**Conflicts of Interest:** The authors declare no conflict of interest.

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
