# Peer review of "Extending the Usage of Newton’s Method with Applications to the Solution of Bratu’s Equation"

_mathematics, doi:10.3390/math6120274_

Reviewer 1 Report

The title and the content of the article is not clear to me. As far as I understand an improvement of the conditions of the Kantorovich theorem is presented, with weaker conditions that ensure the convergence of Newton's method. This is a theoretical study on the convergence of Newton's method.

But this study appears mixed with the issues concerning the Bratu problem. The idea to get an approximate solution of Bratu's problem lies in the use of the Gauss-Legendre quadrature to approximate the integral that appears in the problem. In this way the problem is reduced to solving a system of equations, which is then solved using Newton's method.

The exposure of the contents is not clear. Issues about the result on the convergence of Newton's method and the resolution of Bratu's problem are being mixed.

In my opinion the article should be restructured, presenting the result about the modification of Kantorovich's Theorem, and then, among the examples considered, it may be included Bratu’s problem.

Author Response

RESPONSE TO THE REVIEWER 1

Extending the usage of Newton’s method with applications to the solution of Bratu’s equation

by I. K. Argyros and D. González

November 13, 2018

The title has been changed.

Examples are provided in Section 2 and Section 3.

The semilocal convergence of Newton’s method is now given in Section 2.

Bratu’s equation is treated in Section 3.

All results on Bratu’s equation have been moved to Section 3, whereas Section 2 only contains the new semilocal convergence results by Newton’s method.

We would like to express our gratitude to this reviewer for the constructive criticism of this paper.

I. K. Argyros and D. González

Reviewer 2 Report

The paper applies Kantorovich condition and generalizes the existence of a solution of Bratu's equation. This solution is then guaranteed to be found by the Newton's method. The paper is technically sound, well-written, and provided by examples that help very much. I have only a few minor comments that can be fixed easily.
There are alternative approaches for the root-finding problem than Newton's method. There are global solvers that guarantee to find all roots within the interval, see e.g.
Sederberg, T. W., & Nishita, T. (1990). Curve intersection using Bézier clipping. Computer-Aided Design, 22(9), 538-549.
Bartoň, M., & Jüttler, B. (2007). Computing roots of polynomials by quadratic clipping. Computer Aided Geometric Design, 24(3), 125-141.
where the latter one moreover has cubic convergence rate, while Newton is only quadratic.
My another comment points to the numerical integration part, where m-point Gauss-Legendre formula is used. For problem where the solution is sufficiently smooth, one can use generalized Gauss quadrature rules for splines. That is, instead of projecting f into a space of higher-degree polynomials, one can project it to a spline space, see e.g.
Hiemstra, R., Calabro, F., Schillinger, D.and Hughes, T. J. R., 2017. Optimal and reduced quadrature rules for tensor product and hierarchically refined splines in isogeometric analysis. Computer Methods in Applied Mechanics and Engineering 316, 966-1004.
Johannessen, K., 2017. Optimal quadrature for univariate and tensor product splines. Computer Methods in Applied Mechanics and Engineering 316, 84-99.
Barton, M., Ait-Haddou, R., Calo, V. M., 2017. Gaussian quadrature rules for C1 quintic splines with uniform knot vectors. Journal of Computational and Applied Mathematics 322, 57-70.
These alternative quadratures will most likely not affect the convergence rate, however, they can make the computation more efficient as fewer quadrature points are needed to achieve the same accuracy.
In example 6, the vector \overline(x)_0 is not defiend. Is it a vector that holds the iteration? If so, say it explicitly.
typos and minor issues:-page 5, line 51: convergente -> convergence,-page 7, line 65: we have this -> we have that this,-page 8, line 75: as auxiliary -> as an auxiliary,-page 8, line 78: two "dots",-page 9, line 84: no-decreasing -> non-decreasing,-page 11, line 108: extra dot after "<1",-page line="" 139:="" and="" an="" least="" -=""> and at least.p, li { white-space: pre-wrap; }

Author Response

RESPONSE TO THE REVIEWER 2

Extending the usage of Newton's method with applications to the solution of Bratu's equation

by I. K. Argyros and D. González

November 13, 2018

The title has been changed as suggestion of a reviewer.

We have corrected the following typos:

page 5, line 51: convergente -> convergence

page 7, line 65: we have this -> we have that this

page 8, line 75: as auxiliary -> as an auxiliary

page 8, line 78: two "dots"

page 9, line 84: no-decreasing -> non-decreasing

page 11, line 108: extra dot after "<1"< p="">

page 13, line 139: and an least -> and at least

References have been cited as you requested.

We would like to express our gratitude to this reviewer for the constructive criticism of this paper.

I. K. Argyros and D. González

Reviewer 3 Report

This is a really interesting paper in which authors are concerned with the problem of solving Bratu's equation by means of using the well-known Newton's method. The new criteria beats the old one and as a consequence the pplicability of the method is incrased. Moreover, authors present numerical examples validating the theoretical results. Finally, the paper fits the scope so I strongly recommend the paper for publication in Mathematics.

Author Response

RESPONSE TO THE REVIEWER 3

Extending the usage of Newton's method with applications to the solution of Bratu's equation

by I. K. Argyros and D. González

November 13, 2018

The title has been changed as suggestion of a reviewer.

Thank you.

We would like to express our gratitude to this reviewer for the constructive criticism of this paper.

I. K. Argyros and D. González

Round  2

Reviewer 1 Report

In my opinion the paper has gained readability, and may be considered for publication. There are some minor issues concerning the English that should be addressed:

-          The first sentence in the Abstract is incomplete. Please modify it.

-          After line 11, please correct: To provide the [semilical] semilocal convergence

-          The sentence in line 13 is incomplete. Please, correct it: This way the new convergence analysis shall be at least as precise.